# Improvement in Central Neck Dissection Quality in Thyroid Cancer by Use of Tissue Autofluorescence

**DOI:** 10.3390/cancers16020258

**Published:** 2024-01-06

**Authors:** Octavian Constantin Neagoe, Mihaela Ionică

**Affiliations:** 1Second Clinic of General Surgery and Surgical Oncology, Emergency Clinical Municipal Hospital Timișoara, 300079 Timișoara, Romania; neagoe.octavian@umft.ro; 2Second Discipline of Surgical Semiology, First Department of Surgery, ”Victor Babeș” University of Medicine, 300041 Timișoara, Romania

**Keywords:** thyroid cancers, thyroidectomy, nodal metastasis, autofluorescence

## Abstract

**Simple Summary:**

With the increasing incidence of thyroid cancer, ensuring a safe surgical treatment is essential. For well-differentiated thyroid cancers, total thyroidectomy with central neck dissection is the most common performed surgery. The most frequent postoperative complication is represented by transient or permanent hypocalcemia, as a result of parathyroid gland injury or accidental excision. Autofluorescence is a relatively new and non-invasive method for the intraoperative detection of parathyroid glands, being based on the innate property of parathyroid tissue to become autofluorescent in the near-infrared spectrum. In this study, we show how this technique can significantly decrease the risk of postoperative hypoparathyroidism and improve lymphatic clearance.

**Abstract:**

Background: Risk of postoperative transient or permanent hypoparathyroidism represents one of the most common complications following total thyroidectomy. This risk increases if a cervical lymphadenectomy procedure must also be performed, as is usually the case in thyroid carcinoma patients. Parathyroid autofluorescence (AF) is a non-invasive method that aids intraoperative identification of parathyroid glands. Methods: In this prospective study, 189 patients with papillary thyroid cancer who underwent total thyroidectomy with central neck dissection were included. Patients were randomly allocated to one of two groups: NAF (no AF, surgery was performed without AF) and the AF group (surgery was performed with AF—Fluobeam LX system, Fluoptics, Grenoble, France). Results: The number of excised lymph nodes was significantly higher in the AF compared to the NAF group, with mean values of 21.3 ± 4.8 and 9.2 ± 4.1, respectively. Furthermore, a significantly higher number of metastatic lymph nodes were observed in the AF group. Transient hypocalcemia recorded significantly lower rates in the AF group with 4.9% compared to 16.8% in the NAF group. Conclusions: AF use during total thyroidectomy with central neck dissection for papillary thyroid carcinoma patients, decreased the rate of iatrogenic parathyroid gland lesions, and increased the rate of lymphatic clearance.

## 1. Introduction

The surgery of well-differentiated thyroid cancers serves to increase survival, decrease recurrence rates, and, very importantly, to correctly stage the disease [1]—the aspect on which the future therapeutic plan is established [2,3]. The surgical approach addresses not only the excision of the thyroid gland but also the lymph nodes of the cervical groups, as the lymph nodes represent the most common site of disease persistence and recurrence [1,4,5,6]. At the moment of diagnosis, papillary carcinoma presents lymphatic macrometastasis in a large proportion of cases (50–60%) and up to 90% of cases present micrometastasis [7].

The problem that an extensive central neck dissection poses is that of an increased rate of postoperative complications. As such, if for a total thyroidectomy the mean risk of iatrogenic hypoparathyroidism is 30% (9–27.3% for transient and 2.6–12.1% for permanent hypoparathyroidism, respectively) [8,9,10], for a surgical procedure that also comprises the lymphatic dissection of the central neck compartment, the mean risk is significantly higher, reaching a mean value of 50% (36–68% for transient and 2–18% for permanent hypoparathyroidism, respectively) [9,11,12,13], with a relatively constant risk of recurrent laryngeal nerve injury of 1–2% using intraoperative neurostimulation and neuromonitoring systems [14,15].

Introduced relatively recently in current medical practice, the method of parathyroid detection through autofluorescence in the near-infrared (NIR) spectrum and the intraoperative evaluation of their viability has allowed for a significant reduction in iatrogenic lesions during thyroid surgery, down to a value of 5.2% [16].

The possibility of confirming parathyroid gland localization in the surgical field, the evaluation of their presence on excised tissue through autofluorescence, and the possibility of evaluation, both before and after resection of the vascularization and viability of parathyroid glands by means of indocyanine green (ICG) angiography in NIR spectrum, have made this method the main instrument in combating the fear of producing iatrogenic parathyroid lesions following thyroidectomy [17,18,19,20].

The purpose of the present study was to analyze the effect of parathyroid gland detection through autofluorescence in the NIR spectrum during total thyroidectomy with central neck dissection in patients with papillary carcinoma. Both lymphatic resection efficiency and iatrogenic risk were recorded.

## 2. Materials and Methods

This was a single-center prospective study performed between January 2020 and March 2023, in a high-volume center for thyroid cancer. The study was approved by our Institutional Review Board and informed consent was obtained from all patients. A total of 198 patients diagnosed with papillary thyroid carcinoma were included in the study. Inclusion criteria comprised patients aged over 18 years, with papillary thyroid carcinoma confirmed through fine-needle aspiration biopsy (FNAB) cytology or pathological examination; patients underwent total thyroidectomy with central neck compartment with curative intent (Figure 1). Exclusion criteria comprised incomplete data; papillary thyroid carcinoma that was not histopathologically confirmed or other forms of thyroid cancer (follicular, medullary, anaplastic, etc.); patients with reinterventions following previous thyroid or neck surgery; surgical procedures for thyroid cancer recurrence; surgical procedures with palliative intent; patients with multi-visceral resections (tracheal, esophageal, muscular, and vascular resections); and cases that required extemporaneous histopathological organ confirmation (Figure 1). The following data regarding patient characteristics were collected: age, gender, tumor size (classified according to TNM staging), presence of multifocal thyroid tumors, number of excised lymph nodes from the central neck compartment, number of metastatic lymph nodes from the central neck compartment, number of identified parathyroid glands, number of excised parathyroid glands, and postoperative complications. Hypocalcemia was defined as transient if it was present for less than 6 months postoperatively.

All patients underwent total thyroidectomy with central neck dissection, performed by 2 surgeons with over 7 and 15 years of experience in thyroid and parathyroid surgery (78 patients were operated on by the first surgeon and 120 patients were operated on by the second surgeon, respectively). Patients were allocated randomly to one of two groups: the first group was constituted of patients in which parathyroid gland identification was performed through macroscopic exploration, designated as the NAF or control group, whereas the second group comprised patients in whom autofluorescence was used for intraoperative parathyroid detection, designated as the AF group. Group allocation of patients was performed based on a computer generated random-numbers table. Surgical procedures were performed under general anesthesia with orotracheal intubation using EMG tubes for NIM 3.0 neuromonitoring system by Medtronic (Minneapolis, MN, USA). Patients were placed in a supine position with cervico-thoracic extension. A Kocher type incision was performed about 2 cm above the sternal notch, followed by the incision of the linea alba and the dissection of thyroid lobes by a lateral to medial approach. Central neck dissection was performed through blunt and sharp dissection, with the excision of supraistmic (prelaryngeal), infraistmic, and bilateral paratracheal lymph node groups, starting from the cranial limit (paralaryngeal) to the caudal pole at the level of the brachiocephalic plane. Parathyroid gland identification was performed through macroscopic evaluation alone in the control group; whereas in the study group, macroscopic evaluation was aided through autofluorescence detection in the NIR spectrum (820 nm) with the Fluobeam^®^ LX system (Fluoptics, Grenoble, France).

Statistical analysis was performed with SPSS version 19.0 software (IBM Corporation, Armonk, NY, USA). Analysis of continuous data was performed by an independent sample *t* test, whereas categorical variables were analyzed by a chi-square (χ2) test or Fisher exact test to identify the features associated with use of autofluorescence. All performed tests were two-sided and a *p* value of <0.05 was considered statistically significant.

## 3. Results

Demographic data of the study group can be seen in Table 1. No statistically significant difference was observed for the mean age between the study groups. Gender distribution showed a predominance of female patients in both groups.

With regard to primary thyroid tumor size, almost half of the control group presented T1 lesion (49.5%), while most of the AF group recorded T2 tumors (54.4%); however, this difference was not statistically significant. Only a small proportion of patients in both groups were observed to have T3 primary thyroid tumors. Multifocal papillary carcinoma was recorded in less than 20% of the control group patients, while only 10.7% of the patients in the AF group presented multifocal disease.

Following central neck dissection, a significantly higher number of lymph nodes was observed in the AF group (<0.01). A mean number of 9.2 ± 4.1 nodes were recorded in the control group, while the AF group observed a mean value of 21.3 ± 4.8 removed lymph nodes. Nodal metastasis was present in a larger proportion of the patients in the AF group compared to the control group, with 57.3% compared to 35.8%, respectively (<0.01). Among patients presenting with nodal involvement, a higher number of metastatic lymph nodes were recorded in the AF group compared to the control group, with 8.2 ± 4.4 compared to 3.7 ± 1.5 nodes, respectively (<0.01).

Macroscopic examination and parathyroid gland autofluorescence were used as an intraoperative means for identification and confirmation of the presence and location of parathyroid glands in the AF group. In the control group, only macroscopic evaluation was used for the identification of parathyroid glands. The autofluorescence allowed for a significantly improved identification of parathyroid glands (*p* < 0.01). A minimum of three parathyroid glands were identified in over 93% of the patients from the AF group. By comparison, a minimum of three parathyroid glands were observed in only 77.9% of patients from the control group. As presented in Table 2, all four glands were identified in the majority of patients from the AF group, while in the control group, these were observed in less than half the cases. Supernumerary parathyroid glands were observed to a greater extent in the AF group compared to the NAF group, with a fifth or sixth gland being identified in 10 and 2 patients, respectively. Autofluorescence decreased the number of cases where two or less parathyroid glands were identified, from 22.1% in the control group to 6.8% in the AF group.

Accidental excision of parathyroid glands occurred in almost 18% of patients from the control group, with a significantly lower number of cases being recorded in the AF group (7.8%). The mean number of excised parathyroid glands was similar in both groups (Table 2). In a small number of patients, parathyroid gland excision was intentional due to either preoperatively recorded pathology or intraoperatively identified parathyroid gland disease. Only one parathyroid gland was excised in all patients from this group, with no significant differences between study groups.

Transient hypocalcemia was observed in a significantly higher proportion in the NAF group, with 16.8% (16 patients) in the NAF group compared to 4.9% (5 patients) in the AF group (*p* < 0.01). No significant difference was observed between the groups of patients with hypocalcemia with regard to the mean value of postoperative calcium, recorded at 7.4 ± 0.8 mg/dL in the NAF group and at 7.2 ± 0.4 mg/dL in the AF group, respectively. No patient required more than 7 days of oral calcium supplementation. No cases of permanent hypocalcemia were recorded.

## 4. Discussion

The prognosis of papillary thyroid carcinoma patients depends on a series of factors that include the patient being in a risk class with regard to post-therapeutic disease recurrence, speed of disease progression, and disease-specific survival. Intrinsic factors such as genetic mutations (BRAF and TERT), aggressive histologic subtypes (tall cell, hobnoid variant, and columnar cell carcinoma), presence of lymph-vascular invasion, or multifocal disease in association with demographic factors represented by size, extrathyroid extension of primary tumor, size and number of metastatic lymph nodes, presence of systemic metastasis, resectability of residual iodine captant sites, or increased postoperative thyroglobulin values are elements that comprise the stratification of risk classes [21,22].

Regarding lymphatic infiltration, the presence of clinically or ultrasound metastatic lymph nodes, with or without cytologic confirmation, imposes a selective lymph dissection of the respective compartment. It must be mentioned that 35% of papillary carcinomas present cN1 disease [23,24,25,26], with 36.3% of those with cN1 central compartment metastasis having bilateral nodal involvement of the central neck compartment. Furthermore, 83% of those with latero-cervical adenopathies also present central neck compartment dissemination [27].

As such, for papillary thyroid carcinoma cN1 patients, the central neck dissection is mandatory due to oncologic risk; specific risk factors in this sense are represented by T3-4 tumors, multifocality, extracapsular extension, and, of course, the presence of cervical adenopathies [28,29,30].

The iatrogenic risk following central neck dissection remains relatively constant (1–2%) with regard to superior and recurrent laryngeal nerve injuries [14,15], without significant differences between high- and low-volume centers as long as intraoperative neuromonitoring systems/methods are used.

However, high-volume centers have shown a reduction of 75%, from 16.1% to 4.3% with regard to parathyroid gland lesions. Thus, a decrease from 33–68% to 14–40% for transient postoperative hypocalcemia and from 2–18% to 1.2% for permanent hypoparathyroidism, respectively, represents a significant improvement in morbidity rate; however, it remains at an uncomfortably high level from a cost-benefit point of view.

Studies showing that micrometastasis have a lesser influence in survival than macrometastasis in papillary thyroid carcinoma [31,32,33] have changed the initially aggressive attitude towards prophylactic central neck dissection. Prophylactic central neck dissection that was shown to reduce postoperative thyroglobulin levels [30,34], decrease local recurrence risk [29,34], indicate RAI use [2,3,34,35,36], estimate the risk of recurrence [3,37,38], and improve survival [39], giving in to the fear of iatrogenic lesions. ”Therefore, we need not subject a large majority of patients […] to an operation that increases the risks of recurrent nerve injury and especially parathyroid (PTH) gland injury” [40] is a defensive attitude in sight of a potentially high morbidity in the case of a neoplastic patient.

The number of affected lymph nodes in thyroid carcinoma is of great pathologic, therapeutic, and prognostic importance. However, this aspect may be underappreciated due to the impossibility of performing a correct lymphadenectomy, consequently to the fear of high iatrogenic risks comprising predominantly parathyroid gland lesions.

Surgical approaches such as unilateral lymphatic dissection with or without [41] contralateral lymphatic sampling do not have an oncologic substrate but rather a defensive one in the light of incorrect management of iatrogenic risks; much like the fact that lymph node sampling is not an accepted procedure in cervical cancer. A compromise between a less aggressive approach and a radical lymphatic dissection remains the central neck dissection technique [42,43,44,45,46].

Starting from the premise that the mean number of peritracheal lymph nodes varies between 3 and 30 [47] and that the mean number of excised lymph nodes in thyroid cancer averages 8.4 [42], an appropriate histologic evaluation can be performed by both improving the accuracy of pathologic examination and by increasing the number of excised lymph nodes [23,34]. In our study, we have found that a greater extent of lymphatic dissection could be performed with the use of autofluorescence. This observation is most likely explained by the possibility of accurately identifying parathyroid glands and preserving them, thus allowing for a more radical excision of lymphatic tissue in the central compartment. Furthermore, the increase in excised lymph nodes also associated a significantly higher rate of metastatic lymphatic spread, at 57.3% of patients in the AF group compared to 35.8% in the NAF group. The increase in the mean number of excised lymph nodes could also account for the observed increase in the mean number of metastatic lymph nodes, as pathologists have the possibility to examine a larger tissue sample.

The use of tissue autofluorescence in the NIR spectrum for intraoperative parathyroid gland identification and confirmation has proven to play an important role in significantly reducing the rate of secondary postoperative hypoparathyroidism in the surgery of benign thyroid disease from 20.9% to 5.2% [16]. Autofluorescence made an entrance in thyroid and parathyroid surgery in 2011 as a quantitative method and since 2014 as a qualitative measure [48,49]. This imaging technique is based on the property of parathyroid parenchyma to emit an autofluorescent signal in the 750–785 nm wavelength following previous excitation in the 800–950 nm wavelength. Although this autofluorescent signal is not exclusive to the parathyroid glands—as it may also be seen in certain conditions in thyroid, thymic tissue, white and brown fat, or metastatic lymph nodes—it is dominant in parathyroid parenchyma [50].

One of the systems currently available for parathyroid gland detection through autofluorescence, the Fluobeam system, was shown to present a sensitivity of 98.8% [51,52]. The advantage of this method is represented by the fact that an intensity of the parathyroid glands of 2.4 to 8.5 times greater than surrounding tissues [49] makes it possible to detect parathyroid glands in 37–68% of cases before dissecting them and macroscopic identification [16,53,54].

The use of autofluorescence for the detection or confirmation of parathyroid glands is extremely important [55]. An inherent risk of accidental excision of 8–19% [56,57,58] is due to anatomical variability in up to 16% of cases, with ectopic, subcapsular, or intraparenchimatous locations that determine a macroscopic detection rate of 33% to 63% of parathyroid glands in the operating field [59,60].

With regard to central neck dissection, this procedure increases the risk of iatrogenic lesions of the parathyroid gland due to the macroscopic similarities between tissues located in the paratracheal spaces, namely lymph nodes, white and brown adipose tissue, thymic tissue, and accessory thyroid lobules. Thus, parathyroid gland detection through autofluorescence at the moment of lymphatic dissection can significantly reduce the accidental excision rate [16,61,62,63,64]. Furthermore, autofluorescence offers the possibility of examining excised specimens (thyroid gland and peritracheal tissue) for the detection of accidentally excised parathyroid glands, with the possibility of reimplantation [62,65,66]. Nevertheless, as mentioned earlier, this method can present with limitations due to the risk of false-positives. When excited by NIR light, other tissues may fluoresce, an aspect that bears a particular significance in the context of thyroid cancer. Metastatic lymph nodes have been shown to sometimes present an autofluorescent signal. The ability to distinguish these lymph nodes from parathyroid glands represents an important aspect for the radical surgical treatment of patients with thyroid papillary carcinoma. Although experienced surgeons may differentiate the two types of structures based on visual inspection, this is mostly a subjective evaluation and therefore more prone to error. Intraoperative frozen sections can confirm the nature of the examined tissue and aid in the decision-making process. Additionally, a quantitative measurement of autofluorescence intensity could substitute this need and offer a quick and easy means for reducing the false-positive rate. Such systems are currently being researched; most recently, Makovac et al. proposed a camera-based system, the EleVision IR, for expressing the intensity of autofluorescence as a percentage relative to the surrounding tissue [67].

The second factor that contributes to postoperative hypoparathyroidism is represented by ischemia due to parathyroid vessel damage. Parathyroid gland vascularization can be evaluated through NIR angiography with intravenous administration of ICG. Parathyroid glands with an absent signal, suggesting tissue infarction, can be reimplanted, increasing functionality rate to 75–90% from 30% in the case of in situ abandonment [19,20,51,68].

Limitations of the present study must be acknowledged; parathyroid gland perfusion was not evaluated through ICG angiography, due to the fact that this method was only available for a part of the study group. Another limitation of the current study is represented by the lack of parathormone dosing in the postoperative setting, especially for patients with hypocalcemia. Furthermore, a limitation of the study is represented by the design, that did not meet all the criteria of a clinical trial. Future studies, comprising larger samples and with a RCT design, are necessary to confirm these findings.

## 5. Conclusions

Autofluorescence represents an efficient non-invasive method for parathyroid gland identification that not only allows a decrease in the postoperative hypoparathyroidism rate but may also increase the effectiveness of central neck dissection in thyroid cancer patients. The use of parathyroid gland identification through autofluorescence has significantly increased the mean number of excised lymph nodes from the central neck compartment, with a consequent increase in the number of identified metastatic nodes. Through this method, a more radical surgical treatment of thyroid cancer can be performed, with implications on subsequent risk classification and the therapeutic and follow-up approaches.

## Figures and Tables

**Figure 1 cancers-16-00258-f001:**
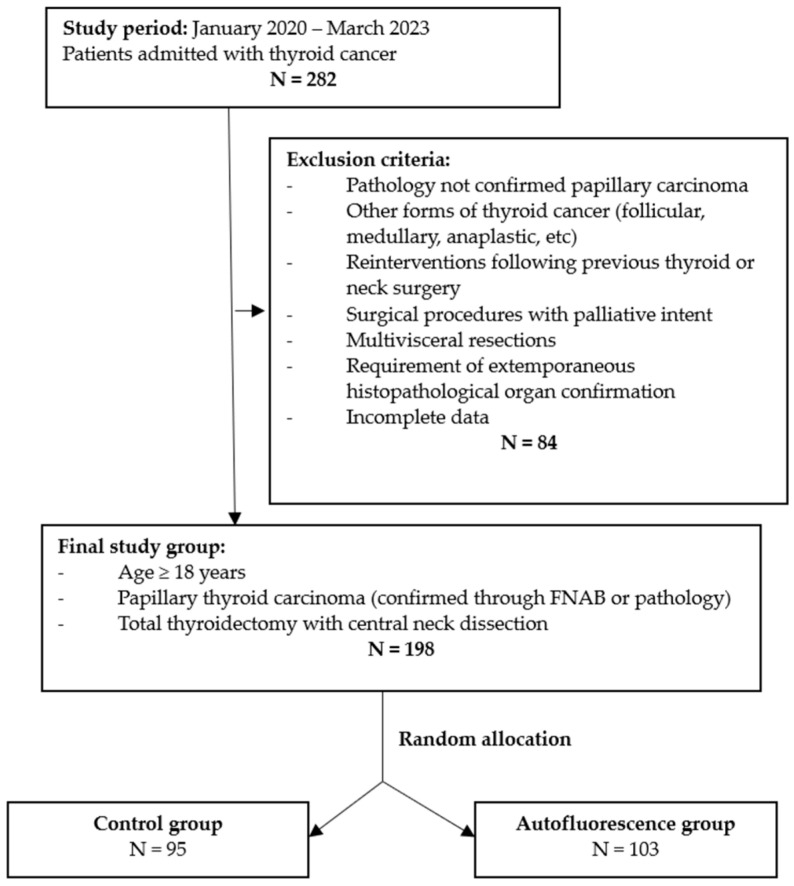
Selection criteria for patients included in the study.

**Table 1 cancers-16-00258-t001:** Distribution of patient and tumor characteristics.

Patient Characteristics	NAF Group N = 95	AF Group N = 103	*p* Value
**Age** (years)	49.7 ± 18.2	50.1 ± 19.1	n.s.
**Gender**			n.s.
Male	17 (17.9%)	21 (20.4%)	
Female	78 (82.1%)	82 (79.6%)	
**Multifocal thyroid papillary carcinoma**			n.s.
Yes	18 (18.9%)	11 (10.7%)	
No	77 (81.1%)	92 (89.3%)	
**Tumor size**			n.s.
T1	47 (49.5%)	38 (36.9%)	
T2	40 (42.1%)	56 (54.4%)	
T3	8 (8.4%)	9 (8.7%)	
**No. of excised lymph nodes**	9.2 ± 4.1	21.3 ± 4.8	<0.01
**Nodal metastasis**			<0.01
Yes	34 (35.8%)	59 (57.3%)	
No	61 (64.2%)	44 (42.7%)	
**No. of metastatic lymph nodes**	3.7 ± 1.5	8.2 ± 4.4	<0.01

NAF group—no autofluorescence group (patients from the control group in whom macroscopic exploration was used for parathyroid gland detection); AF group—autofluorescence group (patients in whom the NIR autofluorescence technique was used for parathyroid gland detection).

**Table 2 cancers-16-00258-t002:** Parathyroid gland identification.

Parathyroid Gland Identification Characteristics	NAF Group N = 95	AF Group N = 103	*p* Value
**No. of identified parathyroid glands**			<0.01
1	5 (5.3%)	2 (1.9%)	
2	16 (16.8%)	5 (4.9%)	
3	28 (29.5%)	21 (20.3%)	
4	44 (46.3%)	65 (63.1%)	
≥5	2 (2.1%)	10 (9.7%)	
**Accidentally excised parathyroid glands**			
Patients	17 (17.9%)	8 (7.8%)	<0.04
Mean no.	1.2 ± 0.4	1.1 ± 0.3	n.s.
**Intentionally excised parathyroid glands**			
Patients	7 (7.4%)	6 (5.8%)	n.s.
Mean no.	1 ± 0	1 ± 0	n.s.

NAF group—no autofluorescence group (patients from the control group in whom macroscopic exploration was used for parathyroid gland detection); AF group—autofluorescence group (patients in whom the NIR autofluorescence technique was used for parathyroid gland detection).

## Data Availability

Data are available on request.

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
