# Peer review of "Improvement in Central Neck Dissection Quality in Thyroid Cancer by Use of Tissue Autofluorescence"

_cancers, 2024, doi:10.3390/cancers16020258_

Round 1

Reviewer 1 Report

Comments and Suggestions for Authors

Dear Editor,

I read with interest the article “Improvement of central neck dissection quality in thyroid cancer by use of tissue autofluorescence.”

This is a prospective study of the usefulness of NIRAF in thyroidectomy + central ND for papillary thyroid cancer.

The study is interesting, the data interesting and well analyzed. The study shows that with NIRAF, more parathyroids are detected and preserved in situ, apparently leading to a reduced risk of post-operative hypocalcemia. Interestingly, the study shows that there are more lymph nodes (and more invaded LN) in the NIRAF group. The authors should discuss this point a little more. What is their opinion on the suspected mechanism? central ND practiced more extensively thanks to the safety of NIRAF for parathyroid preservation? Different surgeon? Different timing?

The authors mention that this is a prospective study, but they do not mention the method of creating the 2 groups (time series? by surgeon? by chance? probably not randomized otherwise it would be mentioned?). The discussion does not mention this important limitation.

Postoperative calcium and PTH data are missing, as is the authors' definition of transient hypocalcaemia (and do the authors have data on definitive hypocalcaemia ?)

It would be interesting to have some information on the possible false-positives of the NIRAF technique on N+ lymph nodes, as mentioned e.g. in the study “Intraoperative Near-infrared Imaging for Parathyroid Gland Identification by Auto-fluorescence: A Feasibility Study. De Leeuw F, Breuskin I, Abbaci M, Casiraghi O, Mirghani H, Ben Lakhdar A, Laplace-Builhé C, Hartl D. (ref 51 of this paper) and “Preliminary experience with the EleVision IR system in detection of parathyroid glands autofluorescence and perfusion assessment with ICG”. Makovac P, Muradbegovic M, Mathieson T, Demarchi MS, Triponez F. Front Endocrinol (Lausanne). 2022 Oct 17;13:1030007. doi: 10.3389/fendo.2022.1030007. eCollection 2022

Author Response

We would like to thank the editors and reviewers for their time and valuable comments, and for offering us the opportunity to improve our work. We have replied to the reviewers questions and have highlighted in yellow changes made in the manuscript.

R1

I read with interest the article “Improvement of central neck dissection quality in thyroid cancer by use of tissue autofluorescence.”

This is a prospective study of the usefulness of NIRAF in thyroidectomy + central ND for papillary thyroid cancer.

The study is interesting, the data interesting and well analyzed. The study shows that with NIRAF, more parathyroids are detected and preserved in situ, apparently leading to a reduced risk of post-operative hypocalcemia. Interestingly, the study shows that there are more lymph nodes (and more invaded LN) in the NIRAF group. The authors should discuss this point a little more. What is their opinion on the suspected mechanism? central ND practiced more extensively thanks to the safety of NIRAF for parathyroid preservation? Different surgeon? Different timing?

A: Thank you for your observations. As you suggested, we have added a paragraph in text to better highlight this aspect. Yes, in our study we have observed that with the use of NIRAF parathyroids are more easily identified, allowing for a safe and more extensive lymphatic dissection to be performed. All surgeries included in this study have been performed by the same team of surgeons (namely, the authors of this paper) – this is mentioned in the methods section. No significant difference in time interval has been observed between the groups.

The authors mention that this is a prospective study, but they do not mention the method of creating the 2 groups (time series? by surgeon? by chance? probably not randomized otherwise it would be mentioned?). The discussion does not mention this important limitation.

A: Thank you for this remark. We have not detailed the method for randomization. Allocation to one of the groups was based on a random-numbers table generated by a computer. This allocation was recorded in an envelope for each patient and this envelope was opened right before the surgery, when the patient was brought in the operating room. This has been added to the methods section of the manuscript.

Postoperative calcium and PTH data are missing, as is the authors' definition of transient hypocalcaemia (and do the authors have data on definitive hypocalcaemia ?)

A: Unfortunatelly, not all patients included in the study had more than a year of follow-up at the end of the study. However, no patients received more than 7 postoperative days of calcium supplementation and at the end of the study, none of the endocrinologists that monitor these patients have reported any cases of definitive hypocalcemia. Data on mean calcium values have been added to the manuscript. PTH was not routinely determined in patients that presented without hypocalcemia symptoms.

It would be interesting to have some information on the possible false-positives of the NIRAF technique on N+ lymph nodes, as mentioned e.g. in the study “Intraoperative Near-infrared Imaging for Parathyroid Gland Identification by Auto-fluorescence: A Feasibility Study. De Leeuw F, Breuskin I, Abbaci M, Casiraghi O, Mirghani H, Ben Lakhdar A, Laplace-Builhé C, Hartl D. (ref 51 of this paper) and “Preliminary experience with the EleVision IR system in detection of parathyroid glands autofluorescence and perfusion assessment with ICG”. Makovac P, Muradbegovic M, Mathieson T, Demarchi MS, Triponez F. Front Endocrinol (Lausanne). 2022 Oct 17;13:1030007. doi: 10.3389/fendo.2022.1030007. eCollection 2022

A: Thank you for this suggestion. As you have mentioned this is an interesting aspect of NIRAF imaging, especially in the context of radical surgical treatment for thyroid cancer. We have added the second study to our reference list and added also a comment on this aspect in the manuscript.

Reviewer 2 Report

Comments and Suggestions for Authors

The authors presented a series of papillary thyroid carcinoma patients treated with total thyroidectomy and central neck dissection with the aid of tissue autofluorescence for the detection of parathyroids gland. The study is interesting: I have some remarks. Were there any differences in term of recurrence between the two groups? Although the group of patients treated with autofluorescence had a lower incidence of complications, however, a more extensive dissection could be justified only if a better oncologic outcome could be obtained. Moreover, was the autofluorence perfomed after the neck dissection? This could increase the rate of parathyroid autotransplantation, further decreasing the rate of postoperative hypoparathyroidism. 

Comments on the Quality of English Language

English language is good

Author Response

We would like to thank the editors and reviewers for their time and valuable comments, and for offering us the opportunity to improve our work. We have replied to the reviewers questions and have highlighted in yellow changes made in the manuscript. 

R2

The authors presented a series of papillary thyroid carcinoma patients treated with total thyroidectomy and central neck dissection with the aid of tissue autofluorescence for the detection of parathyroids gland. The study is interesting: I have some remarks. Were there any differences in term of recurrence between the two groups? Although the group of patients treated with autofluorescence had a lower incidence of complications, however, a more extensive dissection could be justified only if a better oncologic outcome could be obtained.

A: Thank you for your interesting remarks. At the end of the study period we had only 1 patient that was diagnosed with latero-cervical adenopathy within 2 months from the moment of initial surgery. As this was very close to the moment of diagnosis we did not consider this to be a recurrence, but rather a progression of disease. Local or systemic recurrence was not recorded in any patient. The extent of the lymphadenectomy was greater in the AF group mostly due to the use of autofluorescence, as it was easier to correctly identify the parathyroid glands. Increasing the number or excised lymph nodes is also the most likely mechanism that determined an increase in the identification of more metastatic lymph nodes.  Comments on this aspect were added to the discussions section of the manuscript.

Moreover, was the autofluorence perfomed after the neck dissection? This could increase the rate of parathyroid autotransplantation, further decreasing the rate of postoperative hypoparathyroidism.

A: Thank you for this observation. The autofluorescence was performed both during and after the neck dissection. During the procedure it helped us differentiate parathyroid glands from adipose and lymphatic tissue, increasing in situ preservation. After performing the thyroidectomy and central neck dissection, autofluorescence was used to check resection specimens for accidentaly excised parathyroid glands, indeed increasing autotransplantation procedures when required.

Round 2

Reviewer 1 Report

Comments and Suggestions for Authors

If it is indeed a randomized study, the study should be reported according to the CONSORT guidelines.

Author Response

We would like to thank the editors and reviewers for allowing us to answer your questions and further improve our work. We have replied to the reviewers questions and have highlighted in pink changes made in the manuscript for this second revision.

R1

If it is indeed a randomized study, the study should be reported according to the CONSORT guidelines.

A: Thank you for your observation. We apologize if our phrasing was confusing and unclear. All data in this study was collected prospectively, the information mentioned in the methods section with regard to patient data are the variables that we considered for analysis. Although patients were allocated randomly to a group, we did not record, nor consider this study as a clinical trial. This representing the reason for which CONSORT guidelines were not mentioned. We have included this aspect in the paragraph regarding the limitations of our study. In addition, in the limitations paragraph we have mentioned the lack of PTH measurement. The number of surgeries performed by each surgeon was added, as well as the definition of transient hypocalcemia.

We appreciate your understanding and commitment to ensuring the accuracy and transparency of our research.

Reviewer 2 Report

Comments and Suggestions for Authors

the authors addressed all my comments

Author Response

Thank you for your time and valuable recommendations!

Round 3

Reviewer 1 Report

Comments and Suggestions for Authors

The authors answered the questions raised